# Efficacy of N-Acetylcysteine on Endometriosis-Related Pain, Size Reduction of Ovarian Endometriomas, and Fertility Outcomes

**DOI:** 10.3390/ijerph20064686

**Published:** 2023-03-07

**Authors:** Emanuela Anastasi, Sara Scaramuzzino, Maria Federica Viscardi, Valentina Viggiani, Maria Grazia Piccioni, Laura Cacciamani, Lucia Merlino, Antonio Angeloni, Ludovico Muzii, Maria Grazia Porpora

**Affiliations:** 1Department of Experimental Medicine, Sapienza University of Rome, 00161 Rome, Italy; 2Department of Maternal and Child Health and Urological Sciences, Policlinico Umberto I, Sapienza University of Rome, 00161 Rome, Italy; 3Department of Molecular Medicine, “Sapienza” University of Rome, 00161 Rome, Italy

**Keywords:** endometriosis, N-acetylcysteine, NAC, pelvic pain, infertility, pregnancy

## Abstract

Background: Endometriosis is a chronic, estrogen-dependent, inflammatory disease, whose pivotal symptoms are dysmenorrhea, dyspareunia, and chronic pelvic pain (CPP). Besides the usual medical treatments, recent evidence suggests there are potential benefits of oral N-acetylcysteine (NAC) on endometriotic lesions and pain. The primary objective of this prospective single-cohort study was to confirm the effectiveness of NAC in reducing endometriosis-related pain and the size of ovarian endometriomas. The secondary objective was to assess if NAC may play a role in improving fertility and reducing the Ca125 serum levels. Methods: Patients aged between 18–45 years old with a clinical/histological diagnosis of endometriosis and no current hormonal treatment or pregnancy were included in the study. All patients received quarterly oral NAC 600 mg, 3 tablets/day for 3 consecutive days of the week for 3 months. At baseline and after 3 months, dysmenorrhea, dyspareunia and CPP were assessed using the Visual Analog Scale score (VAS), while the size of the endometriomas was estimated through a transvaginal ultrasound. Analgesics (NSAIDs) intake, the serum levels of Ca125 and the desire for pregnancy were also investigated. Finally, the pregnancy rate of patients with reproductive desire was evaluated. Results: One hundred and twenty patients were recruited. The intensity of dysmenorrhea, dyspareunia and CPP significantly improved (*p* < 0.0001). The use of NSAIDs (*p* = 0.001), the size of the endometriomas (*p* < 0.0001) and the serum levels of Ca125 (*p* < 0.0001) significantly decreased. Among the 52 patients with reproductive desire, 39 successfully achieved pregnancy within 6 months of starting therapy (*p* = 0.001). Conclusions: Oral NAC improves endometriosis-related pain and the size of endometriomas. Furthermore, it decreases Ca125 serum levels and may improve fertility in patients with endometriosis.

## 1. Introduction

Endometriosis is a chronic, estrogen-dependent, benign, inflammatory disease, affecting between 2–10% of women in childbearing age, characterized by the presence of endometrial glands and stroma outside the uterine cavity [1]. The pivotal symptom of endometriosis is pain, presenting as dysmenorrhea, dyspareunia, or chronic pelvic pain (CPP), and it is associated with infertility in up to 50% of cases [2,3]. Environmental, dietary, genetic, and immune factors seem to be involved in the etiopathogenesis of endometriosis [4,5,6,7,8,9,10,11]. Above all, the exposure to non-persistent endocrine disruptors, such as dioxins, polychlorinated biphenyls, bisphenol A and phthalates, seems to increase the risk of developing endometriosis, due to the effects on both the endocrine and the immune systems, and to a different capacity in bioactivation and/or detoxication due to both genetic makeup and/or induction/inhibition phenomena in the exposed population [9,10]. The immune system in turn plays a key role in the pathogenesis of endometriosis; growth, angiogenic and adhesion factors, along with proinflammatory cytokines, reside in both the peritoneal fluid and in the endometrium of these patients, enhancing a leading role of inflammation in the initiation and progression of the disease [12]. The inflammatory response in the peritoneal cavity provokes the release of reactive oxygen species (ROS). The ROS include free radicals and non-free-radical oxygen intermediates, which can act on the cellular components and DNA, inducing damage [13]. In physiological conditions, the ROS and the antioxidants are in balance. In case of the ROS overbearing, cellular damage is triggered, causing the downregulation of protein activity and gene expression [14], with subsequent inflammation [15]. Oxidative stress induces the upregulation of these molecules, and it is responsible for the local destruction of the peritoneal mesothelium, for creating adhesion sites for the ectopic endometrial cells, and for promoting apoptosis [16]. The ROS are also implicated in the pathogenesis of endometriosis-related infertility. In fact, the damage produced by the ROS may contribute to a reduced oocyte quality and impaired ovulation [17]. Therefore, an antioxidant diet and the administration of antioxidant drugs may be complementary treatments that go alongside the hormone therapy. 

N-acetylcysteine (NAC) is a substance that exerts antiproliferative and antioxidant effects on tissues. It facilitates the proliferation-to-differentiation switch and downregulates the gene and inflammatory proteins expression [18]. NAC is a precursor of the antioxidant glutathione, with well-established antioxidant and anti-inflammatory properties, due to both a direct and indirect effect. The direct effect relates to the presence of a free thiol group which scavenges the ROS, while the indirect effect involves its ability to enter the cells and to react with glutamic acid and glycine, increasing the levels of free glutathione which reduces the ROS [19]. 

In previous studies, we demonstrated that in both animal and human tissues NAC causes a significant reduction in the size of endometriotic lesions and improves pain symptoms [20]. Due to its strong anti-inflammatory action, NAC may also reduce Cancer Antigen 125 (Ca125) production and improve fertility. 

The objective of this prospective observational single-cohort study was to confirm the effectiveness of N-acetylcysteine in reducing endometriosis-related pain and the size of ovarian endometriomas and to assess a potential role in the reduction of Ca125 serum levels and improvement fertility.

## 2. Materials and Methods

Between January 2020 and April 2022, patients with endometriosis referred to the Endometriosis outpatient service of Policlinico Umberto I -Sapienza University Hospital, were enrolled in this prospective single-cohort study. All recruited patients signed an informed consent to the study. The study was approved by the local Ethic Committee (n. 5926/2020).

Inclusion criteria were age between 18 and 45 years old and clinical/instrumental or surgical/histological diagnosis of endometriosis. Exclusion criteria were pre-menarche or menopause, known hypersensitivity or previous adverse reaction to N-acetylcysteine, current hormonal treatment, cancer and ongoing pregnancy. Age, body mass index (BMI), parity, comorbidities, previous surgery, previous medical treatment, intake of nonsteroidal anti-inflammatory drugs (NSADs), size of ovarian endometriomas, presence of dysmenorrhea, dyspareunia and CPP and desire for pregnancy were investigated. Quarterly therapy with 600 mg oral N-acetylcysteine (3 tablets/day, for 3 consecutive days of the week) was prescribed to all patients for 3 months. Dysmenorrhea, dyspareunia, non-cyclic CPP, size of ovarian endometriomas, and Ca125 levels were evaluated at the beginning of the treatment (t0) and after 3 months (t3). The size of ovarian endometriomas was estimated by a transvaginal ultrasound scan (TVUS) performed by the same expert operator (GE Voluson E6, transvaginal 6 MHz volume probe with 3D scan, GE Healthcare, Milwaukee, WI, USA). Pain symptoms were assessed through the Visual Analogical Scale (VAS), consisting of a 10-point score defining pain as mild (VAS 1–4), moderate (VAS 5–7) and severe (VAS 8–10). Furthermore, serum levels of Ca125 were measured on the 8th day of the menstrual cycle in every patient. CA125 levels were determined using the CA125 non-competitive, indirect, two-step sandwich chemiluminescent immunoenzymatic (CLEIA) method conducted on the LUMIPULSE^®^ G1200 automated analyzer (Fujirebio Diagnostics). According to the manufacturer’s indications, normal values of CA125 were considered to be <35 U/mL. Lastly, fertility outcomes among patients with reproductive desire were evaluated 6 months after starting the therapy, assessing the occurrence of pregnancy, the potential history of infertility, the spontaneous pregnancy rate, and the pregnancy rate after Assisted Reproduction Techniques (ART).

Statistical analysis was performed using SPSS, version 26 for iMac (IBM, SPSS Statistics, Bologna, Italy) provided by “Sapienza” University of Rome.

A preliminary descriptive analysis was performed to assess the patients’ general characteristics. The Shapiro—Wilk test was applied to test for a normal distribution. According to the normal or not-normal distribution, continuous variables were compared using the paired T-test or the Wilcoxon test, as appropriate. Categorical variables were compared using the McNemar test. Statistical significance was set at a *p*-value < 0.05.

## 3. Results

One hundred and twenty patients were included in the study. The average age was 33.2 ± 6.7 years old. The mean BMI was 22.2 ± 3.9. Thirty-four patients (28.3%) had a history of previous surgery for endometriosis. One hundred and four patients (86.6%) had received hormonal treatment before the recruitment, 92 (88.5%) with combined oral contraceptives (COC) or oral progestins, 3 (2.9%) with GnRH analogues and 9 (8.6%) with an intrauterine device releasing Levonorgestrel. Thirty-nine patients (32.5%) had already had at least one pregnancy before the recruitment. The main characteristics of the study population are reported in Table 1.

At t0, 97 patients (80.3%) reported dysmenorrhea, with an average VAS score of 6.9 ± 2.0. At t3, dysmenorrhea was still present in 84 patients (70%, *p* = 0.001) but the mean VAS score was 4.8 ± 1.8 (*p* < 0.0001). Dyspareunia was present in 56 patients (46.7%) at t0 and in 50 patients (41.7%) at t3 (*p* > 0.05) with a significant reduction of mean VAS score, 6.5 ± 1.7 at t0 vs. 4.9 ± 1.7 at t3 (*p* < 0.001). CPP was reported by 50 patients (41.7%) at t0 and by 45 patients (37.5%) at t3 (*p* > 0.05), with a significant improvement of mean VAS score, 7.2 ± 1.8 at t0 vs. 5.7 ± 2.0 at t3 (*p* < 0.001). NSAIDs intake also reduced from 63.3% (*n* = 76) to 53.3% (*n* = 64, *p* = 0.001).

A statistically significant reduction of the endometriomas’ average size was observed, changing from 36.5 mm ± 25.4 mm at t0 to 33.0 mm ± 23.5 mm at t3 (*p* < 0.001). 

The Ca125 average serum levels significantly decreased from 45.55 U/mL ± 26.5 U/mL at t0 to 35.6 U/mL ± 24.2 U/mL at t3 (*p* = 0.001). 

The main results of the study are reported in Table 2.

Fifty-two patients (43.3%) had a pregnancy desire, and twenty-three patients (19.1%) reported a history of infertility. Among the fifty-two patients with reproductive desire, thirty-nine (75%) had a spontaneous pregnancy within 6 months (*p* = 0.001), while six (11.5%) successfully achieved pregnancy through ART, four through intracytoplasmic sperm injection (ICSI) and two through in vitro fertilization (IVF) (Table 3).

Although it was not an aim of the study, we observed a significant decrease in the BMI, as it reduced from 22.2 kg/m^2^ ± 3.9 kg/m^2^ at t0 to 21.2 kg/m^2^ ± 3.8 kg/m^2^ at t3 (*p* = 0.03). No side effects were observed.

## 4. Discussion

Endometriosis is a chronic disease, influenced by environmental, genetic, and epigenetic factors. Some polymorphisms, such as altered Toll-like receptor 4 (TLR4), which is involved in the activation of immune and inflammation responses, seem to be associated with an increased risk of developing the disease [6]. Furthermore, environmental pollutants and endocrine disruptors may play a role by interfering with the endocrine and immune systems and with ROS generation [9,10]. Therefore, due to the involvement of several mechanisms, the treatment of endometriosis is still challenging. Although there is no definitive cure for endometriosis, hormonal treatments are generally prescribed, and have been shown to be effective on both the disease and painful symptoms. However, these therapies may carry side effects and are not indicated in women with a current desire for pregnancy. Patients with endometriosis require a personalized therapy which considers their needs and the overall clinical picture, including the burden of painful symptoms, the extent of the pelvic anatomy impairment and short- or long-term reproductive desire. In patients seeking pregnancy, the use of NAC can be proposed. In fact, NAC proved to be effective and safe, both in vitro and in vivo, in reducing endometriotic lesions and endometriosis-related pain, and it can be safely used in pregnancy, due to the near absence of reported side effects. The decision to administer NAC quarterly, with a total of 9 doses/week, lies in its pharmacokinetic properties, as a reduction in the absorption and blood concentration was reported after prolonged daily treatments with both high and low doses. Furthermore, considering the plasma NAC half-life (less than 3 h), the fractionation into 3 doses of 600 mg each ensured constant plasma levels, as reported in previous studies [21,22]. The purpose of our study was to demonstrate the benefits of NAC in patients with endometriosis. As previously reported in a pilot study that we conducted a few years ago [20], we found a significant improvement in endometriosis-associated pain after 3 months of treatment. In particular, lower mean VAS scores of dysmenorrhea, dyspareunia and CPP were observed, leading to a significant decrease in analgesics intake. The effect on pain is probably due to the strong antioxidant effect of NAC. Oxidative stress and inflammation play a key role in the pathogenesis of endometriosis-related pain. The release of cysteine, a precursor of reduced glutathione, from NAC, allows the molecule to perform an indirect antioxidant action, with the removal of ROS leading to the inhibition of proinflammatory cytokines (IL-6, IL-8, TNF-alpha), VEGF, and metalloproteinases, whose concentration is increased in the peritoneum of patients with endometriosis [15,22]. These results are in line with data reported in a recent review by Mohiuddin et al., which explains the effects of NAC on chronic pain in adults [19]. Moreover, NAC seems to reduce ferroptosis, a mechanism recently demonstrated in endometriotic cells [23]. In the case of retrograde menstruation, the blood present in the peritoneum undergoes erythrocyte degradation. This results in the release of free iron, increased transferrin saturation and the potential intracellular accumulation of ferritin. Iron excess can activate Fenton’s reaction, with a further release of ROS [24]. Our study showed a significant reduction of the size of ovarian endometriomas after three months of treatment. This effect seems to be related to the strong antiproliferative effect of NAC, which causes morphological, biochemical, and molecular changes that lead to a shift from proliferation to cell differentiation. One of the mechanisms involved is related to the inhibition of the tyrosine-kinase c-Src, whose action regulates and influences the adhesive migratory capacities of the cell. It was observed that NAC treatment may induce colon and ovary cancer cells to show an increase in adhesion complexes, linked to the decreased activity of the c-Src, thus stimulating cell differentiation with reduction of proliferation [25]. A study conducted in 2002 reported how NAC may play a role in cell differentiation, demonstrated by the delocalization of E-cadherin from the cytoplasm to the cellular membrane. In fact, patients treated with NAC showed an increased concentration of E-cadherin at the cell-to-cell lines, with cell differentiation induction [26]. 

Our study also showed that, after the administration of NAC, there was a significant reduction in the serum levels of Ca125, a marker which increases in several conditions of peritoneal inflammation. Ca125 is a component of the glycoprotein in the epithelia of celomatic origin, such as the endometrium, the endocervix, the fallopian tubes, the pleura, the pericardium, and the peritoneum. It was also found in the pancreas, the colon, the gallbladder, the stomach, the lungs, and the kidneys [27,28]. This glycoprotein is often used as a marker for ovarian cancer, but it is also altered in endometriosis. The increase in this marker concentration may be due to the proliferation of endometrial cells. Since the serum concentration of Ca125 is elevated in multiple conditions, its role in the diagnosis of endometriosis is significantly reduced, in consideration of its low sensitivity and specificity. Nevertheless, for research purposes, we decided to measure the serum marker concentration in patients affected by endometriosis, and we found a significant reduction after treatment with NAC, probably due to the anti-inflammatory action of the molecule at the peritoneal level. The threshold value used was 35 U/mL. To avoid any bias resulting from the monthly fluctuations of the marker, the blood samples were drawn on the 8th day of the menstrual cycle from all patients. 

Of the fifty-two patients with reproductive desire, thirty-nine successfully achieved spontaneous pregnancy within 6 months from the beginning of therapy, while six achieved pregnancies after ART. The efficacy on fertility outcomes could be related to many factors. Firstly, its undisputed antioxidant efficacy, supported by the improvement of oocyte quality due to the reduced expression of COX-2 [29]. In fact, it was reported that a pro-inflammatory and ROS-dominated environment mediates changes in gametic DNA and, through the stimulation of apoptosis, it promotes embryo fragmentation, implantation failure and abnormalities of the placentation process, significantly increasing the rate of miscarriages and recurrent pregnancy losses [30,31]. Moreover, the mucolytic and fluidifying effect can improve both the quality of cervical mucus, facilitating the ascent of spermatozoa through the female genital tracts, and the tubal function in patient fallopian tubes [32], detectable by using the sonohysterosalpingography [33]. Recently, a study conducted by Fan et al. demonstrated how NAC, in vitro, reduces mitochondrial damage in oocytes [34]. Finally, the reduction of painful symptoms allowed an increased rate of sexual intercourses, with a higher probability of conception. 

Although it was not an objective of our study, we observed a significant reduction of the patients’ BMI. However, it should be noted that 86.6% of the patients used hormonal therapy before recruitment. This anamnestic data is fundamental, as it is known that treatment with progestins or estroprogestins may result in weight gain [35]. Despite this, it is possible to assume a beneficial effect of NAC on body weight due to its action on glucose metabolism. Several clinical and experimental studies highlighted the potential effects of NAC as a therapeutic agent for the treatment of insulin-resistant and type-2 diabetes mellitus [36]. A study conducted on healthy men, showed how intravenous perfusion of NAC during peak glycemic improves insulin sensitivity and increases peripheral glucose uptake. Moreover, NAC reduces the oxidation of lipoproteins, improving the absorption of circulating lipids [36]. Our study showed that NAC is a safe, inexpensive and valuable short-term alternative for the treatment of endometriosis-associated pain in patients who cannot or refuse to take hormonal therapy or with the desire for pregnancy, as the current hormonal therapy does not allow for conception. Further studies on a larger number of patients are necessary to confirm our data.

## 5. Conclusions

N-acetylcysteine is effective in reducing endometriosis-related pain symptoms, the size of endometriomas and the serum levels of Ca125. Furthermore, it showed a positive impact on patient fertility. NAC may represent a good therapeutic option for symptomatic women with endometriosis and pregnancy desire.

## Figures and Tables

**Table 1 ijerph-20-04686-t001:** General characteristics of the study population.

Characteristics of the Study Population	*n* (%)
Previous pregnancies	39 (32.5)
Dysmenorrhea	97 (80.8)
Dyspareunia	56 (46.7)
CPP	50 (41.7)
Previous surgery for endometriosis	34 (28.3)
Laparoscopy	23 (67.7)
Laparotomy	11 (32.3)
Endometriomas	120 (100)
Monolateral	102 (85)
Bilateral	18 (15)
Previous hormonal treatment	104 (86.6)
COC/Progestins	92 (88.4)
GnRH-a	3 (2.9)
Other	9 (8.6)
Desire for pregnancy	52 (43.3)
History of infertility	23 (19.1)

**Table 2 ijerph-20-04686-t002:** Differences in pain symptoms, expressed as dysmenorrhea, dyspareunia and chronic pelvic pain, NSAIDs use, size of ovarian endometriomas and Ca125 serum levels at t0 and t3.

Dysmenorrhea			
	**t0**	**t3**	***p*-value**
***n* (%)**	97 (80.3)	84 (70)	0.01
**mean VAS ± SD**	6.9 ± 2.0	4.8 ± 1.8	<0.0001
**Dyspareunia**			
	**t0**	**t3**	***p*-value**
***n* (%)**	50 (41.7)	45 (37.5)	n.s.
**mean VAS ± SD**	6.5 ± 1.7	4.9 ± 1.7	<0.001
**CPP**			
	**t0**	**t3**	***p*-value**
***n* (%)**	50 (41.7)	45 (37.5)	n.s.
**mean VAS ± SD**	7.2 ± 1.8	5.7 ± 2.0	<0.001
**NSAIDs intake**			
	**t0**	**t3**	***p*-value**
***n* (%)**	76 (63.3)	64 (53.3)	0.001
**Size of ovarian endometriomas**			
	**t0**	**t3**	***p*-value**
**mm ± SD**	36.5 ± 25.4	33.0 ± 23.5	<0.001
**Serum levels of Ca125**			
	**t0**	**t3**	***p*-value**
**U/mL ± SD**	45.5 ± 26.5	35.6 ± 24.2	0.001

n.s. = non significant.

**Table 3 ijerph-20-04686-t003:** Fertility outcome within 6 months of starting therapy with NAC.

Fertility Outcomes	*n* (%)
Desire for pregnancy	52 (43.3)
History of infertility	23 (19.1)
Spontaneous pregnancy after 6 months	39 (75)
Pregnancy through ART after 6 months	6 (11.5)

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
