# Peer review of "Efficacy of N-Acetylcysteine on Endometriosis-Related Pain, Size Reduction of Ovarian Endometriomas, and Fertility Outcomes"

_ijerph, 2023, doi:10.3390/ijerph20064686_

Round 1

Reviewer 1 Report

This efficacy study is definitely much needed to move forward with the potential treatment of endometriosis with N-acetylcysteine. Some minor improvements can be made to your manuscript:

1)      Add “size reduction of ovarian endometriomas” to the title, as it is clearly an aim of the investigation.

2)      The safety of NAC was only briefly mentioned (L 158). It should be very important to elaborate on the tolerability of women with endometriosis.

3)      Add some details on how and where the Ca125 measurement was done (L 101, L 102).

4)      The occurrence of pregnancy does not fully define a woman's reproductive desire considering unplanned pregnancy. A woman's reproductive desires can be impacted by a multitude of factors (like those listed in L 139 to L 141). Therefore, L 103 should be re-phrased and/or explained to justify the implied measurement of the reproductive desire.

5)      Your reference 6 was a nice citation on polymorphism in endometriosis of the study cohort. Did you find any race/genetics association for the efficacy of NAC in this investigation? Add your findings to the discussion even if no association is revealed.

6)      Environmental factors like endocrine disruptors may play a role in endometriosis as cited in Ref. 9. Can you expand LL 46-47 to emphasize such environmental factors?

Author Response

This efficacy study is definitely much needed to move forward with the potential treatment of endometriosis with N-acetylcysteine. Some minor improvements can be made to your manuscript

Thank you for appreciating our work.

1)      Add “size reduction of ovarian endometriomas” to the title, as it is clearly an aim of the investigation.

We changed the title in “Efficacy of N-acetylcysteine on endometriosis-related pain, size reduction of ovarian endometriomas, and fertility outcomes”.

2)      The safety of NAC was only briefly mentioned (L 158). It should be very important to elaborate on the tolerability of women with endometriosis.

Thank you for your suggestion. We improved the sentence as follows: “In patients seeking pregnancy, the use of NAC can be proposed. In fact, NAC proved to be effective and safe, both in vitro and in vivo, in reducing endometriotic lesions and endometriosis-related pain, and it can be safely used in pregnancy, due to the near absence of reported side effects”. Moreover, in the following sentence we explain the decision of administrating NAC 600 mg quarterly, which is the recommended dose for women with endometriosis according to previous studies. 

3)      Add some details on how and where the Ca125 measurement was done (L 101, L 102).

We added details on the Ca125 measurements, as follows: “CA125 levels were determined using the CA125 non-competitive, indirect, two-step sandwich chemiluminescent immunoenzymatic (CLEIA) method conducted on LUMIPULSE® G1200 automated analyzer (Fujirebio Diagnostics). According to the manufacturer’s indications, normal values of CA125 were considered to be less than 35 U/mL”.

4)      The occurrence of pregnancy does not fully define a woman's reproductive desire considering unplanned pregnancy. A woman's reproductive desires can be impacted by a multitude of factors (like those listed in L 139 to L 141). Therefore, L 103 should be re-phrased and/or explained to justify the implied measurement of the reproductive desire.

We re-phrased the sentence, as follows: “Lastly, fertility outcomes among patients with reproductive desire were evaluated 6 months after starting the therapy, assessing occurrence of pregnancy, potential history of infertility, spontaneous pregnancy rate, and pregnancy rate after Assisted Reproduction Techniques (ART)”.

5)      Your reference 6 was a nice citation on polymorphism in endometriosis of the study cohort. Did you find any race/genetics association for the efficacy of NAC in this investigation? Add your findings to the discussion even if no association is revealed.

In this cohort study we did not evaluate the presence of polymorphisms, however we added a sentence in the discussion, as follows: ‘Endometriosis is a chronic disease, influenced by environmental, genetic, and epigenetic factors. Some polymorphisms, such altered Toll-like Receptor 4 (TLR4), which is involved in the activation of immune and inflammation responses, seem to be associated to an increased risk of developing the disease [6]’

6)      Environmental factors like endocrine disruptors may play a role in endometriosis as cited in Ref. 9. Can you expand LL 46-47 to emphasize such environmental factors?

In the Introduction, we emphasized the role of environmental factors as follows: “Above all the exposure to non-persistent endocrine disruptors, such as dioxins, polychlorinated biphenyls, bisphenol A and phthalates, seems to increase the risk of developing endometriosis, due to the effects on both endocrine and immune systems, and to a different capacity in bioactivation and/or detoxication due to both genetic makeup and/or induction/inhibition phenomena in exposed population [9, 10].

Moreover, we added in the Discussion: “Furthermore, environmental pollutants and endocrine disruptors may play a role by interfering with the endocrine and immune systems and with ROS generation [9, 10]”

We updated references adding ref. n. 10: Porpora, M.G.; Medda, E.; Abballe, A.; Bolli, S.; De Angelis, I.; di Domenico, A.; Ferro, A.; Ingelido, A.M.; Maggi, A.; Panici, P.B.; De Felip, E. Endometriosis and organochlorinated environmental pollutants: a case-control study on Italian women of reproductive age. Environ Health Perspect 2009, 117, 1070-1075. doi: 10.1289/ehp.0800273.

Reviewer 2 Report

The present manuscript investigated the use of N-acetylcysteine as a therapy for the treatment of endometriosis-associated pain and its efficacy in improving human fertility. The authors demonstrated that the 3 months treatment with N-acetylcysteine was able to decrease the size of endometriomas and presented beneficial effects on painful symptoms and fertility. Overall, the article is well organized with a clear purpose and an objective writing. The discussion addresses all the most relevant points related to human endometriosis and its associated inflammation, pain and fertility.

I thus recommend accepting the manuscript for publication in the current form.

Author Response

Thank you for appreciating our work. We only made minor changes due to the other Reviewers’ request

Reviewer 3 Report

This was an interesting study in which patients with endometriosis received oral N-acetylcysteine (NAC) treatment. It was found that patient symptoms were significantly improved after 3 months of NAC treatment and there was a decrease in the size of endometriomas (via ultrasonography) as well as a significant decrease in serum Ca125 (a non-specific marker of endometrial disease).

This is a well written study with some minor typographical errors, mainly related to numbers (eg ‘hundred-twenty’ should be re-written as one hundred and twenty). In the results section I don’t understand what is meant by ‘4 recurring to intracytoplasmic sperm injection and 2 recurring…’, I don’t think ‘recurring’ is the correct term here. There is mis-spelling of cadherin in the discussion. The sentence ‘Ca125 is a glycoprotein of membrane, present in epithelia…’, do the authors mean ‘Ca125 is a component of the glycoprotein in epithelia…’

Author Response

This was an interesting study in which patients with endometriosis received oral N-acetylcysteine (NAC) treatment. It was found that patient symptoms were significantly improved after 3 months of NAC treatment and there was a decrease in the size of endometriomas (via ultrasonography) as well as a significant decrease in serum Ca125 (a non-specific marker of endometrial disease).

Thank you for the punctual summary of our work.

This is a well written study with some minor typographical errors, mainly related to numbers (eg ‘hundred-twenty’ should be re-written as one hundred and twenty). In the results section I don’t understand what is meant by ‘4 recurring to intracytoplasmic sperm injection and 2 recurring…’, I don’t think ‘recurring’ is the correct term here. There is mis-spelling of cadherin in the discussion. The sentence ‘Ca125 is a glycoprotein of membrane, present in epithelia…’, do the authors mean ‘Ca125 is a component of the glycoprotein in epithelia…’

Thank you for appreciating our work. We checked the spelling mistakes through the help of a native English speaker.